# Conservation laws in quantum noninvasive measurements

Stanisław Sołtan,[1] Mateusz Frączak,[1] Wolfgang Belzig,[2,*] and Adam Bednorz[1]

[1]*Faculty of Physics, University of Warsaw, ul. Pasteura 5, PL02-093 Warsaw, Poland*
[2]*Fachbereich Physik, Universität Konstanz, D-78457 Konstanz, Germany*

(Dated: March 22, 2021)

Conservation principles are essential to describe and quantify dynamical processes in all areas of physics. Classically, a conservation law holds because the description of reality can be considered independent of an observation (measurement). In quantum mechanics, however, invasive observations change quantities drastically, irrespective of any classical conservation law. One may hope to overcome this nonconservation by performing a weak, almost noninvasive measurement. Interestingly, we find that the nonconservation is manifest even in weakly measured correlations if some of the other observables do not commute with the conserved quantity. Our observations show that conservation laws in quantum mechanics should be considered in their specific measurement context. We provide experimentally feasible examples to observe the apparent nonconservation of energy and angular momentum.

## I. INTRODUCTION

Conserved quantities play an important role in both classical and quantum mechanics. According to the classical Noether theorem, the invariance of the dynamics of a system under specific transformations [1] implies the conservation of certain quantities: translation symmetry in time and space results in energy and momentum conservation, respectively, rotational symmetry in angular momentum conservation and gauge invariance in a conserved charge. In quantum mechanics, the observables (in the Heisenberg picture) are time-independent when they commute with the Hamiltonian. Furthermore, some conserved quantities, like the total charge, commute also with all observables. We shall call them *superconserved*. Classically all conserved quantities are also superconserved. In high energy nomenclature the former are known as *on-shell* conserved whereas the latter are called *off-shell* conserved [2]. The concept of superconservation is closely related to the superselection rule which constitutes an additional postulate that the set of observables is restricted to those commuting with the superconserved operators [3].

Conservation principles become less obvious when one tries to verify them experimentally. While an ideal classical measurement will keep the relevant quantities unchanged, neither a nonideal classical nor any quantum measurement will necessarily reflect the conservation exactly. Even the smallest interaction between the system and the measuring device (detector) may involve a transfer of the conserved quantity. The system might become a coherent superposition of states with different values of a conserved quantity (e.g. energy), or in the case of a superconserved quantity an incoherent mixture (e.g. charge). The problem of proper modeling of the measurement of quantities incompatible with conserved ones

has been noticed long ago by Wigner, Araki and Yanase (WAY)[4–6], later been discussed in the context of consistent histories [7], modular values [8], and the quantum clock [9]. The generation, measurement, and control of quantum conserved quantities, in particular, angular momentum, has become interesting recently, both experimentally and theoretically [10–12]. Measurements incompatible with energy lead to thermodynamic cost [13–15].

The quantum objectivity is one aspect of the general concern of Einstein [16] and Mermin [17] if the (quantum) Moon exists when nobody looks. The randomness of quantum mechanics does not exclude objective reality [18]. Here, we assume that objective observations should be noninvasive i.e., leaving the probed system unchanged. Unfortunately, unlike the classical case, the fundamental uncertainty prevents a completely noninvasive measurement in quantum mechanics [19]. Hence, the only remaining possibility seems to be to consider the limit of weak measurements, which are almost noninvasive. The objectivity based on weak measurements can lead to unexpected results such as weak values [20] or the violation of the Leggett-Garg inequality [21–23]. Unlike the standard projection, which is highly invasive, the extraction of objective values from weak measurements requires a special protocol involving the subtraction of a large detection noise. Therefore, such objectivity is debatable [24, 25]. In our opinion, weak quantum measurements are the closest counterparts of classical measurements [26], so they are prime candidates to define objective reality and, consequently, conservation principles are expected to hold in systems with an appropriate invariance.

In this paper, we will show that for quantum measurements in the weak limit superconservation holds but quantities such as energy, momentum, and angular momentum apparently violate conservation even if an appropriate symmetry results in classical conservation law. The violation of conservation appears in third-order time correlations as we illustrate in simple model systems (Fig. 1). The violation is caused by (at least two) other observables that are not commuting with the conserved

*Electronic address: Wolfgang.Belzig@uni-konstanz.de

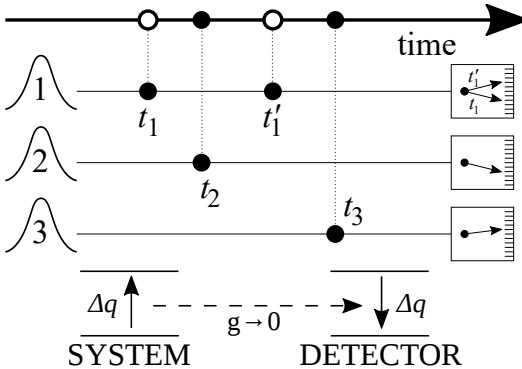

FIG. 1: (top) Three weak measurements. The detectors are initially independent and couple instantly to the system at $t_1$ (or $t_1'$), $t_2$ and $t_3$. The conserved quantity is measured (empty circles) either at time $t_1 < t_2$ or $t_1' > t_2$. The outcomes 1 and $1'$ inferred from the three-point correlator might differ, even for a conserved quantity in the quantum case. (bottom) Failure of conservation in the weak measurement. The transfer $\Delta q$ of the conserved quantity $q$ between the system and detector does not scale with the measurement strength $g$.

one. We write down an operational criterion to witness the violation of a conservation principle and discuss when it is satisfied. Then we propose a feasible experiment probing position and magnetic moment of a charge in a circular trap. Last, considering an imperfect conservation or measurement of the quantity we develop then a Leggett-Garg-type test of objective realism.

## II. SUPERCONSERVATION

The physical Hermitian quantity $\hat{Q}$, defined within the system is *conserved* when $[\hat{H}, \hat{Q}] = 0$ for the system's Hamiltonian $\hat{H}$. The $\hat{Q}$ can be *superconserved* if there exists a set $\mathcal{A}$ of allowed Hermitian observables such $[\hat{A}, \hat{Q}] = 0$ for every $\hat{A} \in \mathcal{A}$. In principle, one could make every conserved quantity superconserved by a proper choice of the set $\mathcal{A}$. However, for instance, for a component of angular momentum we would have to exclude position and momentum or even other components of angular momentum. Instead, we will distinguish quantities that are conserved but not superconserved by allowing measurement of observables not commuting with them. An example of a superconserved quantity is the total electric charge, while the set of observables and possible initial state density matrices is restricted to those that do not change the charge. This is also known as superselection rule. Whether this rule is an axiom or a practical assumption, depending in the considered Hamiltonian, is a matter of debate [3], because one can in principle imagine dynamics without e.g. charge superselection. Nevertheless, here we treat superselection and superconservation as an axiom for certain quantities, like charge. Let us assume the decomposition of a superconserved quantity $\hat{Q} = \sum_q q \hat{P}_q$ where $\hat{P}_q$ are (mutually commuting) projections onto the eigenspace of the value $q$ (i.e. $\hat{Q}|\psi_q\rangle = q|\psi_q\rangle$). Now, the superselection postulate says that the state of the system $\hat{\rho}$ is always an incoherent mixture $\sum_q \hat{P}_q \hat{\rho} \hat{P}_q$, if $\hat{Q}$ is superconserved. Then the projective measurement of $\hat{A}$ will not alter the $q-$eigenspace as there exists a decomposition $\hat{A} = \sum_{q,a} a \hat{P}_{qa}$ with $\hat{P}_{qa}$ being the projection onto the joint eigenspace of $\hat{Q}$ and $\hat{A}$ with respective eigenvalues $q$ and $a$. For instance, if the initial state is already a $q-$eigenstate then it will remain such an eigenstate after the projection. For general measurements, positive operator-valued measures (POVM), represented by Kraus operators $\hat{K}_c$ (the index $c$ can represent an eigenvalue of $\hat{A}$, $\hat{Q}$ or both but in general it can be arbitrary) such that $\sum_c \hat{K}_c^\dagger \hat{K}_c = \hat{1}$, the state $\hat{\rho}$ will collapse to $\hat{\rho}_c = \hat{K}_c \hat{\rho} \hat{K}_c^\dagger$, normalized by the probability $\mathrm{Tr}\hat{\rho}_c$. In principle $\hat{K}_c$ can act within $q$-eigenspaces, i.e. $c = qa$ and $\hat{K}_{qa} = \hat{P}_q \hat{K}_a \hat{P}_q$. In the most general case, the superconserving Kraus operator reads

$$\hat{K}_{q'aq} = \hat{P}_{q'} \hat{K}_a \hat{P}_q. \tag{1}$$

It means that the superconserved value can change but the system remains an incoherent mixture of $q$-eigenstates. This applies e.g. to a charge measurement in a quantum dot (which is superconserved), where the charge can leak out into an incoherent bath. The (normally) conserved quantities do not impose any additional postulates so the state can be a coherent superposition of the states of different values of energy, angular momentum, etc. A projective measurement of $\hat{A}$ which does not commute with $Q$ is enough to turn a $q$-eigenstate into a superposition. Now, if we try to postulate a POVM with superconserving Kraus operators then the actually measured operator involves a linear combination of $\sum_{qq'} \hat{P}_q \hat{K}_a^\dagger \hat{P}_{q'} \hat{K}_a \hat{P}_q$ so it must commute with $\hat{Q}$ which would become superconserved. This is a modern version of the WAY theorem [4–6], that the measurement of $\hat{A} = \sum_a a \hat{K}_a^\dagger \hat{K}_a$ not commuting with $\hat{Q}$, cannot consist of only $\hat{K}_{q'aq}$ defined above with $q-$eigenspaces of $\hat{Q}$. Such a formulation is simpler than the original WAY theorem, as is does not need the dicussion of an auxiliary detector. On the other hand both approaches are equivalent due to the Naimark theorem [27].

The unavoidability of coherent superpositions of only conserved values is the key problem considered here.

## III. WEAK MEASUREMENTS AND OBJECTIVE REALISM

Strong projections are highly invasive, i.e. $\hat{\rho} \rightarrow \sum_P \hat{P} \hat{\rho} \hat{P}$ changes the state $\rho$ very much. On the other hand, unlike classical physics, quantum mechanics does not offer completely noninvasive measurements. The only possibility is weak measurement [20] where we apply Kraus operators

$$\hat{K}_g(a) = (2g/\pi)^{1/4} \exp(-g(\hat{A} - a)^2), \tag{2}$$

with the measurement strength $g \to 0_+$ so that the state almost does not change,

$$\hat{\rho} \to \int da \hat{K}(a) \hat{\rho} \hat{K}(a) \approx \hat{\rho} - g[\hat{A}, [\hat{A}, \hat{\rho}]]/2. \quad (3)$$

The actually measured probability $p'(a) = \mathrm{Tr}\hat{K}(a)\hat{\rho}\hat{K}^\dagger(a)$ of the outcome $a$ at the state $\hat{\rho}$ has a form of convolution

$$p'(a) = \int D(a - A)p(A)dA, \quad (4)$$

$$p(A) = \langle \delta(A - \hat{A}) \rangle = \mathrm{Tr}\delta(A - \hat{A})\hat{\rho},$$

with the dominating detection noise $D(a) = \sqrt{2g/\pi}e^{-2ga^2}$, with $\langle a^2 \rangle_D = 1/4g$, diverging for $g \to 0$. The quantity $p(A)$ is the probability of the outcome $A$ in the case of a strong, projective measurement $g \to \infty$ ($p(A) = \lim_{g\to\infty} p'(A)$), to which the noise $D$ is added. Therefore we can expect that $p(A)$ is in fact the probability that the quantity $\hat{A}$ has objectively the value $A$. Unfortunately such an idea fails in sequential measurements, as shown already in [20], because $p(A, B)$ can be negative when measuring first $\hat{A}$ and then $\hat{B}$, such that $[\hat{A}, \hat{B}] \neq 0$. The original concept [20] involved postselection, i.e. the last measurement is strong, not weak, and the conditional probability $p(A, B)/p(B)$ is considered. However, the strength of the last measurement is irrelevant, as the system is not touched any more. In our approach, all measurements, including the last one, can be assumed weak.

We shall discuss the problem of a negative $p$ in Section VI. Nevertheless, $p(A, B, ...)$ is well defined in the limit $g \to 0$ and we can probe it, hence, assuming that it reflects a property of the system. Note that this construction is still correct in the superconserved case because $\hat{A}$, $\hat{K}_a$ and the state $\hat{\rho}$ is commuting with $\hat{Q}$ so $\hat{K}_a$ splits into a simple sum of $\hat{K}_{qa}$. The actual form of $\hat{K}_a$ can be different but the outcome is almost independent in the limit $g \to 0$. In the lowest order we can also neglect all $\hat{K}_{q'aq}$. In the $g \to 0$ limit, the $n^{\text{th}}$-order correlation of a sequence of measurements $\hat{A}$, $\hat{B}$, $\hat{C}$, $\hat{D}$ with respect to $p$ (or also $p'$ if the quantities are different) reads [26, 28, 29]

$$\langle a(t) \rangle = \langle \hat{A}(t) \rangle, \quad (5)$$

$$\langle a(t_1)b(t_2) \rangle = \langle \{\hat{A}(t_1), \hat{B}(t_2)\} \rangle/2, \quad (6)$$

$$\langle a(t_1)b(t_2)c(t_3) \rangle = \langle \{\hat{A}(t_1), \{\hat{B}(t_2), \hat{C}(t_3)\}\} \rangle/4, \quad (7)$$

$$\langle a(t_1)b(t_2)c(t_3)d(t_4) \rangle$$
$$= \langle \{\hat{A}(t_1), \{\hat{B}(t_2), \{\hat{C}(t_3), \hat{D}(t_4)\}\}\} \rangle/8 \quad (8)$$

for $t_1 < t_2 < t_3 < t_4$ with the anticommutator $\{\hat{A}, \hat{B}\} = \hat{A}\hat{B} + \hat{B}\hat{A}$ and quantum averages $\langle \hat{X} \rangle = \mathrm{Tr}\hat{X}\hat{\rho}$.

At this stage, we would like to note that the classical counterpart of this protocol replaces the anticommutators like $\{\hat{A}, \hat{B}\}$ by simple products of a phase space functions $A(q, p)$ and $B(q, p)$. The invasiveness (3) can be reduced to zero and the time order of observables is irrelevant. For a more detailed analysis of classical-to-quantum correspondence we refer the reader to [26].

## IV. CONSERVATION IN WEAK MEASUREMENTS

The conservation means that the measurable correlations (7) involving the conserved quantity $q(t)$ corresponding to $\hat{Q}(t) = \hat{Q}$ will not depend on $t$. It is true at the single average, where $\langle q(t) \rangle = \langle \hat{Q} \rangle$. Interestingly, also for second order correlations the order of measurements has no influence on the result, since $\langle q(t_1)a(t_2) \rangle = \langle \{Q, A(t_2)\} \rangle/2$ is independent of $t_1$. However, the situation changes for three consecutive measurements (see Fig. 1), since in the last line of (7) the time order of operators matters, which has been demonstrated also experimentally [30]. Considering the difference of two measurement sequences $Q \to A \to B$ and $A \to Q \to B$, we obtain the jump (which is absent in perfectly noninvasive classical measurements [26])

$$\langle \{\hat{Q}, \{\hat{A}(t_2), \hat{B}(t_3)\}\} - \{\hat{A}(t_2), \{\hat{Q}, \hat{B}(t_3)\}\} \rangle = \quad (9)$$
$$\langle [[\hat{Q}, \hat{A}(t_2)], \hat{B}(t_3)] \rangle \equiv 4 \langle \Delta q a(t_2)b(t_3) \rangle.$$

This quantity will show up as jump $\Delta q = q(t_1) - q(t_2)$ at $t_1 = t_2$, when measuring $\langle q(t_1)a(t_2)b(t_3) \rangle$. The jump will be non-zero for $Q$ not commuting with $A$ and $B$. Obviously, for superconserved quantities $Q$ (commuting with every measurable observable) the jump is absent. The violation of the conservation principle is caused by the measurement of $\hat{A}$ – not commuting with $\hat{Q}$ – which allows transitions between spaces of different $q$ with the jump size $\Delta q$ not scaled by the measurement strength $g$, see Fig. 1. This difference is transferred to the detector, assuming that the total quantity (of the system and detector) is conserved regardless of the system-detector interaction. This observation can be compared to the WAY theorem, which applies to projective or general measurements. Here, we have shown that even taking the special limit of noninvasive measurement, the noncommuting quantity causes a jump in third (and higher) correlations. We can call it weak-WAY theorem, as both input (the special construction of $g$−dependent measurements) and the output (correlations) are based on weak measurements. Note that imposing the condition that the jump (9) vanishes, equivalent conservation of $\hat{Q}$ at the level of third-order correlation for an arbitrary state $\hat{\rho}$ (allowed by superselection rules if any apply), namely

$$[[\hat{Q}, \hat{A}], \hat{B}] = 0 \quad (10)$$

for all allowed observables $\hat{A}$ and $\hat{B}$ suffices to keep conservation also at all higher order correlations. Then $\hat{Q}$ is not necessarily superconserved, it can commute with observables to identity, like momentum and position. This subtle difference between the weak-WAY and the traditional WAY theorem in sketched in Fig. 2

As an example we can take the basic two-level system ($|\pm\rangle$ basis) with the Hamiltonian $\hat{H} = \hat{Q} = \hbar\omega|+\rangle\langle+|$ and $\hat{A} = \hat{B} = \hat{X} = |+\rangle\langle-| + |-\rangle\langle+|$. Then, with $\omega > 0$ the ground state is $|-\rangle$ and the third order correlation $\langle h(t_1)x(t_2)x(t_3) \rangle$ for the ground state for $t_3 > t_{1,2}$

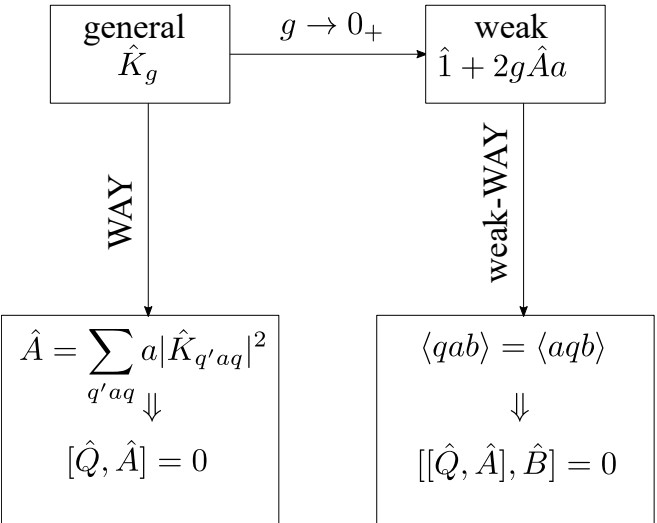

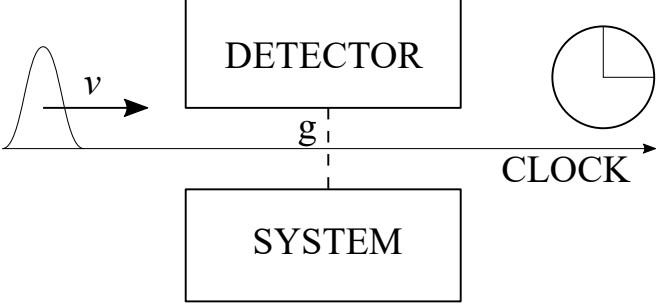

FIG. 4: Detection model based on a clock. The clock is a localized particle traveling with a constant speed $v$. The interaction between the detector and system takes place only when the clock is passing the interaction point.

FIG. 2: The difference between the WAY and the weak-WAY theorem. The former applies to general measurement and shows that lack of coherence between eigenstates of $\hat{Q}$ in Kraus operators (1) leads to superconservation of the measured quantity. The latter applies to weak measurements (2), leading to a weaker condition for the observed conservation.

less position $\sqrt{2}\hat{X} = \hat{a}^\dagger + \hat{a} = \hat{A} = \hat{B}$, we find for the jump $\langle \Delta h x(0)x(t)\rangle = -\hbar\omega\cos(\omega t)/4$ independent of the state of the system (see Appendix A). As illustrated in Fig. 3, the jump becomes unobservable at high temperatures since the average energy $\langle h\rangle = \hbar\omega/[\exp(\hbar\omega/kT)-1]$ increases with temperature.

The previously discussed very simple examples illustrate the fundamental finding of our manuscript. If one tries to verify the conservation of energy while measuring an other observable that is not commuting with the Hamiltonian it is possible to find a violation of the energy conservation. It constitutes a pure quantum effect since it vanishes at high temperature, where the classically expected conservation holds. One could object that performing a series of measurements already breaks time-translational symmetry and therefore the total energy is not conserved. However, one can keep the time symmetry by replacing the detector-system interaction by a clock-based detection scheme [9], see Fig. 4. The total Hamiltonian reads

$$\hat{H} + \hat{H}_x + \hat{H}_z + \hat{H}_I \tag{12}$$

where $\hat{H}$ is the system's part, $\hat{H}_x$ – the detector's part, $\hat{H}_z$ - the clock's part, and finally $\hat{H}_I$ is the interaction between the clock, the system and detector. Each part is time-independent so the time translation symmetry is preserved. Both the detector and the clock can be represented by single real variables, $x$ and $z$. Now, to measure the system's $\hat{A}$ at time $t_1$ we set $\hat{H}_x = 0$ and

$$\hat{H}_z = v\hat{p}_z, \quad \hat{H}_I = g\hat{A}\delta(\hat{z})\hat{p}_x \tag{13}$$

where $\hat{p}_{x,z}$ are conjugate (momenta), i.e. $\hat{p}_x = -i\hbar\partial/\partial x$ and $g \to 0$ is a weak coupling constant. The initial state (at $t = 0$) reads $\hat{\rho}\hat{\rho}_x\hat{\rho}_z$, where both $\hat{\rho}_{x,z} = |\psi_{x,z}\rangle\langle\psi_{x,z}|$ are taken as Gaussian states

$$\psi_z(z) = (2\pi\sigma)^{-1/4}\exp(-(z+vt_1)^2/4\sigma),$$
$$\psi_x(x) = (\pi/2)^{-1/4}\exp(-x^2), \tag{14}$$

respectively. For small $g$ and $\sigma$ the interaction effectively occurs at time $t = t_1$ and, in the end (after the clock

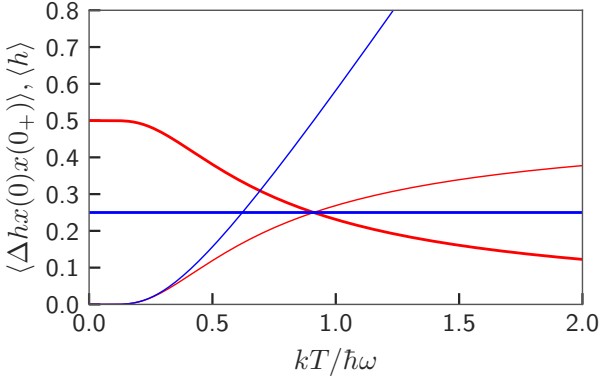

FIG. 3: The non-conserving jump for $\tau = 0_+$ (thick lines) compared to the average energy (thin lines) for the two-level system with level spacing $\hbar\omega$ (red) and the harmonic oscillator with eigen frequency $\omega$ (blue). At high temperatures the jump becomes unobservable and the classical conservation is restored. All quantities are normalized to $\hbar\omega$.

reads $\hbar\omega(1 - \theta(t_2 - t_1))\cos(\omega(t_2 - t_3))/2$. The jump is $\langle \Delta h x(0)x(\tau)\rangle = \hbar\omega\cos(\omega\tau)/2$ for $\Delta h = h(0_-) - h(0_+)$. The result can be generalized to a thermodynamical ensemble with a finite temperature $T$ and reads (see Appendix A)

$$\langle \Delta h x(0)x(\tau)\rangle = \hbar\omega\cos(\omega\tau)\tanh(\hbar\omega/2kT)/2. \tag{11}$$

For increasing temperature the jump diminishes as illustrated in Fig. 3.

Another basic example is the harmonic oscillator with $\hat{H} = \hat{Q} = \hbar\omega\hat{a}^\dagger\hat{a}$ with $[\hat{a}, \hat{a}^\dagger] = 1$. Taking the dimension-

decouples the system and the detector again) to lowest order we find (see details in Appendix B)

$$\langle x \rangle \simeq g \langle \hat{A} \rangle = g \langle a(t_1) \rangle. \qquad (15)$$

For sequential measurements one simply adds more independent detectors and clocks, obtaining in the lowest order of $g$

$$\begin{aligned}
\langle x_A x_B \rangle &\simeq g^2 \langle a(t_1) b(t_2) \rangle, \\
\langle x_A x_B x_C \rangle &\simeq g^3 \langle a(t_1) b(t_2) c(t_3) \rangle, \qquad (16) \\
\langle x_A x_B x_C x_D \rangle &\simeq g^4 \langle a(t_1) b(t_2) c(t_3) d(t_4) \rangle,
\end{aligned}$$

with the right hand sides are given by the quantum expressions (7).

Although the above detection model is based on time-invariant dynamics, the initial state of the clock spoils the symmetry. The time-invariant state would require a constant flow of particles or field a constant velocity, so that the position on the tape imprints time of measurement, see [31] for detailed construction. However, such a constant interaction between the detector (clock) and the system leads to a backaction and makes the measurement invasiveness growing with time, which needs to be reduced by additional resources, e.g. additional coupling to a heat bath.

In order to show that the nonconservation can also occur independent from the time asymmetry present either intrinsically or induced by a quantum clock one can look at other quantities that are conserved, e.g., due to spatial symmetries. As example, we will use one component of the angular momentum in a rotationally invariant system in the following.

## V. ANGULAR MOMENTUM CONSERVATION

We propose an experiment to demonstrate the failure of the conservation principle for angular momentum in third-order correlations in weak measurements. Instead of energy we consider one component of angular momentum, say $\hat{L}_z = \hat{X}\hat{P}_y - \hat{Y}\hat{P}_x$ which can be measured in principle by a sensitive magnetometer (e.g. a superconducting quantum interferometer device). The other two observables will be the particle's positions $\hat{X}$ and $\hat{Y}$, with the readouts $x$ and $y$ respectively, which can be measured e.g. by the voltage of a capacitor depending linearly on $x$ and $y$ for small changes in position, see the setup sketch in Fig. 5. The two positions $x$ and $y$ will be measured at times $t_2$ and $t_3$, respectively.

The quantity of matter is $\langle l_z(t_1) x(t_2) y(t_3) \rangle$. Suppose the particle is in a harmonic trap rotationally invariant about $z$ axis. The $xy$ part of the trap Hamiltonian reads $\hat{H}_\perp = \hbar\omega(\hat{a}_x^\dagger \hat{a}_x + \hat{a}_y^\dagger \hat{a}_y)$, with $[\hat{a}_{x,y}, \hat{a}_{x,y}^\dagger] = 1$, $[\hat{a}_x, \hat{a}_y] = [\hat{a}_x^\dagger, \hat{a}_y] = 0$. Then $\sqrt{2}\hat{X} = \hat{a}_x^\dagger + \hat{a}_x$ and $\sqrt{2}\hat{P}_x/i\hbar = \hat{a}_x^\dagger - \hat{a}_x$ (rescaled by a length unit), similarly for $y$, and $\hat{L}_z = i\hbar(\hat{a}_x \hat{a}_y^\dagger - \hat{a}_y \hat{a}_x^\dagger)$. In the ground state $|0\rangle$ we have $\hat{L}_z|0\rangle = 0$ so only $\langle \hat{Y}(t_3)\hat{L}_z(t_1)\hat{X}(t_2) \rangle$

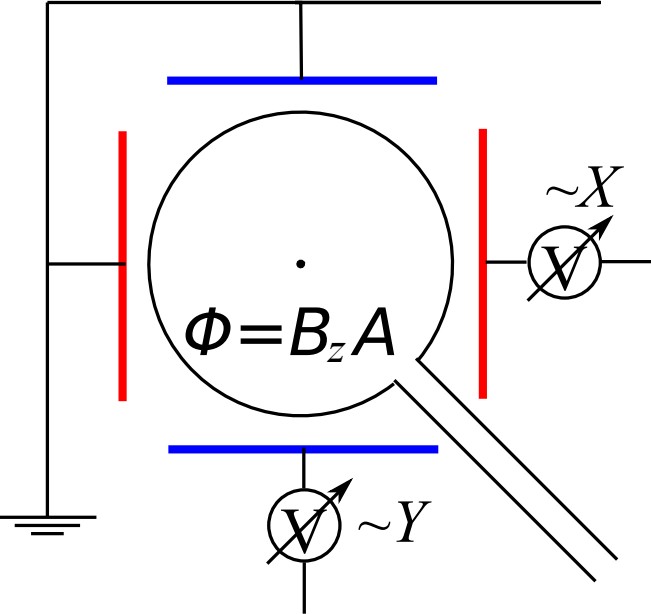

FIG. 5: A trap for a charged particle invariant under rotation about $z$ axis with a detector weakly coupled to angular momentum $L_z$ and the position in the $x-y$ plane detected by appropriate capacitors. The magnetic moment of the particle can be measured by a sensitive magnetometer, e.g. a SQUID

and $\langle \hat{X}(t_2)\hat{L}_z(t_1)\hat{Y}(t_3) \rangle$ contribute in (7). These terms can appear only when $t_2 < t_1$ or $t_3 < t_1$. We find

$$\begin{aligned}
\langle l_z(t_1) x(t_2) y(t_3) \rangle &= (1 - \theta(t_2 - t_1)\theta(t_3 - t_1)) \times (17) \\
\langle \hat{X}(t_2)\hat{L}_z(t_1)\hat{Y}(t_3) &+ \hat{Y}(t_3)\hat{L}_z(t_1)\hat{X}(t_2) \rangle / 4 \\
&= (1 - \theta(t_2 - t_1)\theta(t_3 - t_1))\hbar\sin\omega(t_2 - t_3)/4.
\end{aligned}$$

The jump is therefore given by

$$\langle \Delta l_z x(t_2) y(t_3) \rangle_0 = \hbar\sin\left[\omega(t_2 - t_3)\right]/4 \qquad (18)$$

and is again state-independent as in the case of the harmonic oscillator. It illustrates that the angular momentum conservation is violated by this experiment. At finite temperature $T$ for $t_1 < t_{2,3}$, the correlator $\langle l_z x(t_2) y(t_3) \rangle = \hbar\sin[\omega(t_2 - t_3)]/4\sinh^2(\hbar\omega/2k_B T)$ increases with temperature and makes the (temperature-independent) jump unobservable.

Since in this setup the detectors are coupled permanently, a frequency-domain measurement might be more appropriate. In the frequency domain, the observables are $A(\alpha) = \int dt e^{i\alpha t} A(t)$. Taking all our previous arguments to frequency domain, the conservation of a quantity $\hat{Q}(\alpha)$ means that correlators vanish for $\alpha \neq 0$. Interestingly, transforming to frequency domain we find at zero temperature and for $\gamma, \alpha, \beta \neq 0$ that

$$\langle l_z(\gamma) x(\alpha) y(\beta) \rangle = \frac{i\pi\hbar\omega(\beta - \alpha)\delta(\gamma + \alpha + \beta)}{2(\alpha^2 - \omega^2)(\beta^2 - \omega^2)} \qquad (19)$$

The conservation principle for angular momentum is violated by (19) because it is non-zero. Hence, either by

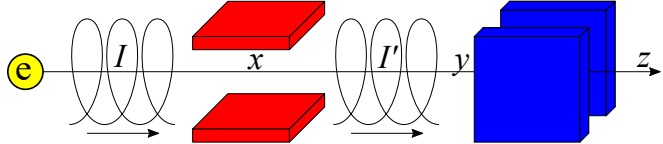

FIG. 6: A charged particle (e.g. electron) goes along the tube with angular momentum measured by current in either of two coils, $I$ or $I'$ while the position in $x$ and $y$ direction is measured by two perpendicular capacitors.

time- or frequency-resolved measurements, one should see experimentally the nonconservation of angular momentum.

To realize a time-resolved measurement, we suggest to test the angular momentum conservation with a charge moving inside a round tube along $z$ direction, similar to the recent test of the order of measurements [30]. In the simplest model take $\hat{H} = \hat{H}_z + \hat{H}_\perp$ and we keep the same harmonic potential in the $xy$ plane as above and add some $\hat{H}_z = v\hat{p}_z$ with velocity $v$ (like the clock form the previous section). Preparing a wavepacket as a product of the ground state of $\hat{H}_\perp$ and $\psi(z)$ of sufficiently short width, we can measure essentially the same quantity (17) by putting a sequence of weak detectors along the tube, see Fig. 6. The angular momentum can be measured by the current signature in the coil, like in the recent experiment [10]. We simplify the coil-electron beam interaction to $\lambda(z)\hat{I}\hat{L}_z$ where $\lambda$ is only non-zero inside the coil. Similarly, the measurement of $x$ and $y$ can be modeled by local capacitive couplings. In this way, the measurement times are translated into position according to $t = z/v$, just like in the time-invariant energy detection in the previous section. The jump (18) can then be detected by placing the coil at two different positions, see Fig. (6).

As regards the rotational invariance of the system, detection of $X$ and $Y$ can be performed by the detector-system coupling

$$\hat{H}_I = g\delta(\hat{z})(\hat{X}\hat{p}_{xD} + \hat{Y}\hat{p}_{yD}) \qquad (20)$$

with the initial state $\hat{\rho}\hat{\rho}_{x,y}^D$ and the detector's state $\hat{\rho}_{x,y}^D = |\psi\rangle\langle\psi|$,

$$\psi(x_D, y_D) = (\pi/2)^{-1/2}\exp(-(x_D^2 + y_D^2)) \qquad (21)$$

Then both the interaction and the initial state of the system and detector are rotationally invariant so the total angular momentum is conserved. Only the readout, either $x$ or $y$ of the detector, prefers one direction, i.e.

$$\langle x_D\rangle \simeq g\langle x\rangle \qquad (22)$$

with straightforward generalization to sequential measurements like (16).

## VI.  LEGGETT-GARG INEQUALITIES

The above proposals face some practical challenges. The velocity $v$ should be sufficiently high in order to ig-

nore decoherence effects, e.g. due to coupling to a thermal environment. The decoherence could be modeled by Lindblad-type terms added to the Hamiltonian dynamics of the density matrix. The test of conservation makes sense only for times/frequencies within the coherence timescale. Any observable roughly tracking the charge in two perpendicular directions will suffice. The tube may be not perfectly harmonic or not homogeneous in the $z$-direction, and $\hat{L}_z$ can be only approximately conserved or imprecisely measured. To quantify these considerations, we will now develop a Leggett-Garg-type test [21]. Let us consider the measurement of four observables: $q = q(t_1)$, $q' = q(t_1')$, $x = x(t_2)$ and $y = y(t_3)$ with $q$ being an *approximate* value of the conserved quantity and $t_1 < t_2 < t_1' < t_3$. Here $q, q'$ can correspond to angular momentum $l_z$, while $x, y$ are the lateral positions in the test presented in the previous section. Note that LG-type tests for angular momentum were discussed in a different context in [33, 34]. According to the objective realism assumption, the values of $(q, q', x, y)$ exist independent of the measurement. If there is a corresponding joint positive probability $p(q, q', x, y)$, then correlations with respect to $p$ must satisfy the following two Cauchy-Bunyakovsky-Schwarz inequalities

$$\langle(q-q')^2\rangle_\rho\langle x^2 y^2\rangle_\rho \geq \langle(q-q')xy\rangle_\rho^2,$$
$$\langle(q-q')^2 y^2\rangle_\rho\langle x^2\rangle_\rho \geq \langle(q-q')xy\rangle_\rho^2. \qquad (23)$$

However if we test these inequalities using $p$ defined in (4 and quantum correlations (7) then they could be violated. Classically, the measurements of the conserved quantity at two different times should not depend on whether another observable is measured in between and both sides of Eqs. (23) vanish. Using Eqs. (7), the left hand sides of Eqs. (23) vanish for a perfectly conserved quantity. First, $\langle(q-q')^2\rangle_p = 0$ because $\hat{Q}(t) = \hat{Q}$ is independent of time. Second, $\langle(q-q')^2 y^2\rangle_p = 0$ because in addition $y$ is measured after both $q$ and $q'$. On the other hand, the right hand side of (23) exactly corresponds to the quantum mechanical jump in the third-order correlator as defined in (18). Hence, even if $q$ is not exactly conserved then the left hand sides can be small enough to violate the inequalities. These violations can be readily tested in the setup suggested in Fig. 6. Note that the inequalities must involve fourth moments because of the so-called weak positivity [32] stating that lower moments are insufficient to violate realism for continuous variables.

## VII.  CONCLUSION

We have shown that conservation laws in quantum mechanics need to be considered with care since their experimental verification might depend on the measurement context even in the limit of weak measurements. The conservation is violated if extracting objective reality from the weak measurements. It means that either (i) weak measurements cannot by considered noninvasive, or

(ii) the conservation laws do not hold in quantum objective realism. Exceptions are superconserved observables, which will be conserved whatever measurement will be performed, and more generally observables, that satisfy the weak-WAY condition (10). The nonconservation can also be formulated as Leggett-Garg-type test showing the connection to the absence of objective realism in quantum mechanics. In the future, it might be interesting on one hand to study more realistic scenarios for quantum measurements taking into account decoherence or more general detectors [35]. Furthermore, one might generalize these findings to more fundamental relativistic field theories [36, 37], testing correlations involving components of stress-energy-momentum tensor.

### Acknowledgements

We thank E. Karimi for fruitful discussion and N. Gisin for bringing the quantum clock to our attention. W.B. gratefully acknowledge the support from Deutsche Forschungsgemeinschaft (DFG, German Research Foundation) through Project-ID 32152442 - SFB 767 and Project-ID 425217212- SFB 1432.

### Appendix A: Derivation of correlation jumps

To derive (11) one needs to take the thermal state

$$\hat{\rho} = Z^{-1}(e^{-\hbar\omega/kT}|+\rangle\langle+| + |-\rangle\langle-|) \qquad (A1)$$

with $Z = 1 + e^{-\hbar\omega/kT}$ and

$$\hat{X}(t) = e^{-i\omega t}|+\rangle\langle-| + e^{i\omega t}|-\rangle\langle+| \qquad (A2)$$

plugged into (9) with $\hat{Q} = \hat{H} = \hbar\omega|+\rangle\langle+|$ and $\hat{A} = \hat{B} = \hat{X}$ with $t_1 = 0_\pm$, $t_2 = 0$ and $t_3 = \tau$.

The case of harmonic oscillator can be written in Fock basis $|n\rangle$, $n = 0, 1, 2, ...$ with $\hat{a}|n\rangle = \sqrt{n}|n-1\rangle$, $[\hat{a}, \hat{a}^\dagger] = 1$, $\hat{n} = \hat{a}^\dagger\hat{a} = \sum_n n|n\rangle\langle n|$, $\hat{n}|n\rangle = n|n\rangle$ and $\hat{H} = \hat{Q} = \hbar\omega\hat{n}$, $\hat{X} = (\hat{a} + \hat{a}^\dagger)/\sqrt{2}$. The thermal state

$$\hat{\rho} = Z^{-1}e^{-\hbar\omega\hat{n}/kT} \qquad (A3)$$

with $Z = (1 - e^{-\hbar\omega/kT})^{-1}$ and

$$\hat{A} = \hat{B} = \hat{X}(t) = (e^{i\omega t}\hat{a} + e^{-i\omega t}\hat{a}^\dagger)/\sqrt{2} \qquad (A4)$$

The independence of the jump of the state follows from the fact that

$$[[\hat{H}, \hat{X}(t_2)], \hat{X}(t_3)] \propto \hat{1} \qquad (A5)$$

because $[\hat{H}, \hat{X}]$ is linear (momentum) in $\hat{a}$ and $\hat{a}^\dagger$ and, hence, the outer commutator becomes a $c$-number.

### Appendix B: Weak correlations with a quantum clock

A single detector and a single clock are defined by (13) and (14), respectively. The detector position $x$ is measured (projectively) at some time later than the interaction moment $t_1$. The average

$$\langle x \rangle \simeq \text{Tr} \int dt[\hat{x}, \hat{H}_I(t)]\hat{\rho}/i\hbar \qquad (B1)$$

in the interaction picture, in the lowest order, according to the decomposition (12). With

$$\hat{H}_I(t) = g\hat{A}(t)\delta(\hat{z} - vt)\hat{p}_x \qquad (B2)$$

for the initial initial states (14) we have used the identity

$$[\hat{C}\hat{D}, \hat{\rho}] = \{C, [\hat{D}, \hat{\rho}]\} + \{\hat{D}, [\hat{C}, \hat{\rho}]\} \qquad (B3)$$

for $\hat{C}\hat{D} = \hat{D}\hat{C}$, $\langle\psi_x|\{\hat{x}, \hat{p}_x\}|\psi_x\rangle = 0$ (anticommutator), $[\hat{x}, \hat{p}] = i\hbar$, $\langle\psi_z|\delta(\hat{z} - vt)|\psi_z\rangle = |\psi_z(vt)|^2$ to get (15). To extend it to the sequential case (16) we do not apply the trace over the system space in (B1) (only over $x$ and $z$), getting the matrix

$$\{\hat{A}(t), \hat{\rho}\}/2 \qquad (B4)$$

Note that it is Hermitian but not positive definite. Nevertheless we can apply the above scheme iteratively, replacing the initial system's state by (B4) to get (16).

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
