# Peer review of "Is energy conserved when nobody looks?"

_SciPost Physics_

## Round 2 · Referee Report · Anonymous (Referee 1) · 2019-12-14

Strengths

1- Timely topic 2- Nicely written 3- Proper references and positioning 4- Scientifically valid

Weaknesses

1- Dangerous (oversold) conclusions

Report

This is an interesting paper that is reminiscent of recent developments of quantum foundations (contextual objectivity) and quantum thermodynamics (measurement driven engines)

However, I always find dangerous that sentences like "Our observation cast some doubt on the compatibility of conservation laws and quantum objectivity", since both are necessary conditions to practice physics.

In the case of energy, energy can indeed change while measuring an observable which does not commute with the Hamiltonian, but this does not violate the world's energy conservation since the measurement channel provides energy.
Actually, the authors summarize this quite well in their conclusion: Conservation laws are contextual.

Requested changes

1- Milden the last sentence of the abstract

---

## Round 3 · Referee Report · Viktoriia Kornich (Referee 2) · 2020-3-25

Strengths

1) Interesting topic. 2) Different models (experiments) are suggested. 3) The authors use short, understandable sentences. The manuscript is indeed written for other people to understand it. 4) The parts of the manuscript are logically ordered and connected.

Weaknesses

1) I have doubts about the validity of experiments. Please, see the report. 2) In case the experiments are valid, the importance of the results seem to be exaggerated. From the general intuition, even the weak perturbation of the system (measurement) can be amplified and make a big impact, depending on a system. However, this is not my main concern.

Report

First, the main questions.

1) In the end of Section 3 the authors state that indeed, the measurements destroy the translational symmetry of time and thus energy might be not conserved. The argument that the other quantity is also not conserved (angular momentum component in this case) does not solve this problem. Maybe there are more reliable arguments for this? They should be added into the manuscript.

2) Both experiments for the z-component of angular momentum use square capacitors, which violate rotational symmetry of space. Then the angular momentum should not be conserved in any case. Is there some explanation for this?

Now, less important questions, aimed mainly at improving the readability of the manuscript.

1) On the page 5, it is written that the magnetic field is produced by the particle itself. Is there some reason for this? Why not simply external magnetic field?

2) In Section 5, the authors use macrorealism assumption. Why? The system under consideration is not macroscopic, why does it necessarily have properties (and why particularly q,q',x,y) which can be determined without perturbing the past or future state of the system?

3) In the end of page 7, why the noise D is called large? For the considered limit q->0 it goes to zero.

Requested changes

Please, clarify all the questions described in the report.

---

## Round 3 · Author Response

Dear Editor,

We are grateful for the comments of the Referee. We agree that the bottom line of our paper is that it follows from quantum mechanics that conservation laws must be contextual.

We have followed the suggestion of the Referee and changed the last sentence of the abstract. In addition we have applied minor improvements of the manuscript detailed below.

Adam Bednorz
on behalf of all the authors

---

## Round 3 · List of Changes

a) Modified the last sentence of the abstract
b) Sightly reordered and modified the paragraph below (8) to emphasize the difference between
classical and quantum case.
c) Minor style changes at the end of Sec. 1 and beginning of Sec. 5.

---

## Round 4 · Referee Report · Viktoriia Kornich (Referee 2) · 2020-5-18

Report

My questions were addressed sufficiently well. I think, the manuscript might be still not very precise, but I agree that it can be published, because it will contribute to the ongoing research in this field.

---

## Round 4 · Author Response

Dear Editors,

we thank for the report and the helpful remarks by the referee. We have revised the manuscript and kindly ask for further consideration. The details of changes and comments are below.

Yours sincerely, Stanisław Sołtan, Mateusz Frączak, Wolfgang Belzig, and Adam Bednorz ----------------------------------------- Reply/Comments to the Report:

Dear Dr. Kornich,

We first would like to thank you for taking the time to assess our manuscript. We also thank for listing the strengths of our work, on which we agree completely. There are some doubts about the validity of the experiments, and we try to clarify them in the detailed response below. We hope we are able to convince you that with the applied changes the work is a valuable addition to the ongoing discussion about the foundations of quantum theory. We are convinced that the finding will trigger further investigations, e.g. in more concrete experimental setting.

Sincerely yours, Stanisław Sołtan, Mateusz Frączak, Wolfgang Belzig, and Adam Bednorz

Detailed response:

Report: 1) In the end of Section 3 the authors state that indeed, the measurements destroy the translational symmetry of time and thus energy might be not conserved. The argument that the other quantity is also not conserved (angular momentum component in this case) does not solve this problem. Maybe there are more reliable arguments for this? They should be added into the manuscript.

Reply: In the previous version we have not discussed the detection process at all, using only Kraus operators. We decided to add a minimal description of the detection mechanics and a new figure 3 which helps to understand the subtleties of the symmetries and detection, at the end of section 3. In particular, one can construct a model based on a so-called quantum clock, where the Hamiltonian is manifestly time-independent.

Report: 2) Both experiments for the z-component of angular momentum use square capacitors, which violate rotational symmetry of space. Then the angular momentum should not be conserved in any case. Is there some explanation for this?

Reply: Indeed, square capacitors break the rotational symmetry, but it has negligible effect on the system due to the weakness of the measurement. In the revised version we added the detector model with interaction preserving total angular momentum. As we explain in section 5, in a real experiment the symmetry leading to conservation will be probably only approximate and, since the measurement should be weak, a slight asymmetry should not matter. In that section, we derive an operational criterion in the form of a Legget-Garg-typ inequality, which can be used to quantify the violation.

Report: 1) On the page 5, it is written that the magnetic field is produced by the particle itself. Is there some reason for this? Why not simply external magnetic field?

Reply: The magnetic field is of Zeeman type, i.e. orbital movement makes the charge an effective magnet. Since it is only technical and might be confusing, we removed this comment from the text. The actual measurement model of angular momentum is explained on page 7, between (12) and (13).

Report: 2) In Section 5, the authors use macrorealism assumption. Why? The system under consideration is not macroscopic, why does it necessarily have properties (and why particularly q,q',x,y) which can be determined without perturbing the past or future state of the system?

Reply: Indeed, it is not "macro". The term "macroscopic realism" has been coined by Leggett and Garg [20] for a hypothetical macrosopic systems (SQUID) maintaining quantum coherence. However, in practice LG simply used the Hamiltonian of a two-level system and the macroscopic realization is irrelevant. In this sense, our considerations would apply to macroscopic quantum systems as well. Nevertheless, we decided to change "macroscopic" to "objective" which is indeed more appropriate in our opinion. Then, the existence of a positively defined probability of q,q',x,y is equivalent to objectivity. Whether quantum mechanics is consistent with such objectivity is a matter of ongoing debate, to which our discussion of conservation laws hopefully contributes.

Report: 3) In the end of page 7, why the noise D is called large? For the considered limit q->0 it goes to zero.

Reply: We thank for spotting this inconsistency. We meant large variance and corrected the text.

---

## Round 4 · List of Changes

1) added a minimal description of the detection mechanics and a new figure 3 2) added the detector model with interaction preserving total angular momentum 3) removed confusing comment from the text about magnetic field 4) changed "macroscopic" to "objective" 5) added "diverging variance" in section 5 Other changes: corrected a few typos corrected and improved the wording in some instances.

---

## Editorial Decision

rejected_or_withdrawn